# The nematode *Caenorhabditis elegans* and the terrestrial isopod *Porcellio scaber* likely interact opportunistically

**Heather Archer** [ID], **Selina Deiparine** [ID], **Erik C. Andersen** [ID]*

Department of Molecular Biosciences, Northwestern University, Evanston, IL, United States of America

* erik.andersen@northwestern.edu

## Abstract

Phoresy is a behavior in which an organism, the phoront, travels from one location to another by 'hitching a ride' on the body of a host as it disperses. Some phoronts are generalists, taking advantage of any available host. Others are specialists and travel only when specific hosts are located using chemical cues to identify and move (chemotax) toward the preferred host. Free-living nematodes, like *Caenorhabditis elegans*, are often found in natural environments that contain terrestrial isopods and other invertebrates. Additionally, the *C. elegans* wild strain PB306 was isolated associated with the isopod *Porcellio scaber*. However, it is currently unclear if *C. elegans* is a phoront of terrestrial isopods, and if so, whether it is a specialist, generalist, or developmental stage-specific combination of both strategies. Because the relevant chemical stimuli might be secreted compounds or volatile odorants, we used different types of chemotaxis assays across diverse extractions of compounds or odorants to test whether *C. elegans* is attracted to *P. scaber*. We show that two different strains–the wild isolate PB306 and the laboratory-adapted strain N2 –are not attracted to *P. scaber* during either the dauer or adult life stages. Our results indicate that *C. elegans* was not attracted to chemical compounds or volatile odorants from *P. scaber*, providing valuable empirical evidence to suggest that any associations between these two species are likely opportunistic rather than specific phoresy.

## Introduction

A phoretic animal, or phoront, hitches a temporary ride on a host in order to disperse to new locations. The relationship between phoront and host is commensal. Phoronts can be generalists using numerous host species or specialists with a single or few specific hosts. Most phoronts are animals that have a limited ability to travel any significant distance by their own power. In order to disperse, they must rely on the movement of a more mobile host, *e.g.* mites traveling via beetles or lice hitching a ride on hippoboscid flies [1,2]. Animals with such low mobility often have a fitness advantage if they disperse to new habitats because dispersal reduces competition for food and/or mates, helps individuals avoid predation, and can facilitate increased gene flow between populations thereby reducing an accumulation of deleterious

**Data Availability Statement:** All relevant data are within the manuscript and its Supporting Information files.

**Funding:** This work was funded by an NSF CAREER Award (1751035) to E.C.A. The funders

had no role in study design, data collection and analysis, decision to publish, or preparation of the manuscript.

**Competing interests:** The authors have declared that no competing interests exist.

mutations and inbreeding depression. Moreover, when populations are dependent on unstable food sources, animals must disperse to find a new food source or populations will starve. Therefore, interactions between phoront and host are critical to species survival.

The nematode *Caenorhabditis elegans* has been isolated in association with several terrestrial invertebrate species, including snails, slugs, and isopods [3]. These associations have assumed to capture phoretic relationships where *C. elegans* is using larger invertebrates as vectors for travel. Consistent with this observation, *C. elegans* dauer larvae appear to seek dispersal vectors using a stage-specific behavior (called nictation) in which individuals stand on their tails, move their bodies in a waving motion, and attach themselves to objects passing nearby such as larger invertebrates [4]. Moreover, genetic differences across natural populations influence this behavior, which suggests that variation in this trait is subject to evolutionary selection [5]. However, it is unknown whether *C. elegans* seeks out specific invertebrates as phoretic hosts or randomly attaches to whatever organism happens to be nearby. Additionally, although dauer larvae are the life stage most frequently found in association with invertebrates, other life stages have been isolated as well [3,6,7] suggesting that phoretic association with invertebrates is not necessarily limited to the dauer stage and to nictation behavior.

*C. elegans* might have specific phoretic hosts similar to what has been observed in other closely related nematode species. For example, phoretic associations between *Pristionchus pacificus* and scarab beetles [8], *Caenorhabditis japonica* and the shield bug *Parastrachia japonensis* [9,10], and the facultative parasite *Phasmarhabditis hermaphrodita* and slugs of the genus *Arion* or *Deroceras* as well as the snail *Helix aspersa* [11,12,13]. An association with slugs and snails has also been observed with *C. elegans*, where nematodes have been recovered from the intestines and feces of *Arion sp.* slugs [14,6]. Moreover, previous studies have shown that mucus of the slug *Arion subfuscus* and the snail *Helix aspersa* act as strong chemoattractants for *P. hermaphrodita* [11,13]. Taken together, these observations suggest that *C. elegans* might also detect and move toward chemical cues from invertebrates that act as hosts for dispersal.

To determine whether phoretic interactions between *C. elegans* and the terrestrial isopod *Porcellio scaber* are facilitated by a chemical stimulus, we tested the chemotactic behaviors of *C. elegans* toward *P. scaber*. Because the relevant chemical stimuli might be secreted compounds or volatile odorants, we used different types of chemotaxis assays across diverse extractions of compounds or odorants. Additionally, we tested two different genetic backgrounds and different developmental stages. Across all these different conditions, our results indicate that *C. elegans* was not attracted to chemical compounds or volatile odorants from the isopod *P. scaber*, providing valuable empirical evidence to suggest that any associations between these two species are likely opportunistic.

## Results

The canonical *C. elegans* strain, N2, has been continuously domesticated in a laboratory environment since its initial isolation from mushroom compost in 1951 [15]. This long-term propagation caused the accumulation of laboratory-derived alleles with associated phenotypic effects, and any research to understand *C. elegans* behavior in the wild must take this potential limitation into consideration. Because N2 has been removed from essentially all natural ecological interactions during its domestication in the lab and raised almost exclusively in association with *Escherichia coli* as a food source, it is likely genetically and behaviorally distinct and does not resemble most of its wild counterparts [16,17,18,19,15]. The C. elegans strain PB306 is a wild isolate collected as dauer juveniles from the body of an isopod (Porcellio scaber) from Connecticut Valley Biological Supply by Scott Baird in 1998 [20], and the geographic origin of the isopods is not known). As such, PB306 is likely to have invertebrate-associated traits intact

and is reasonable to hypothesize that these traits would be observed in interactions with *P. scaber*. For these reasons, we tested both N2 and PB306 for chemoattraction towards *P. scaber* compounds that could be secreted or bound to the surface of the isopod. Because the chemical nature of any potential secretions is unknown, a set of three solvents capable of solubilizing both polar and nonpolar compounds was used: ethanol, dimethyl sulfoxide (DMSO), and deionized water. None of these three solvents elicited any chemotactic behaviors (attraction or repulsion, S1 Fig).

## *C. elegans* adults are not attracted to compounds from the terrestrial isopod *P. scaber*

If *C. elegans* seeks out specific hosts, then one possible hypothesis is that chemicals secreted by the host into the local environment form a basis for seeking behaviors. The terrestrial isopod *Armadillidium vulgare* uses sex-specific short-distance chemical cues for mate attraction [21]. This result suggests that another isopod, *P. scaber*, could also secrete different pheromones between the sexes. For this reason, we treated male and female isopods separately. In order to test whether any secretions act as a chemoattractant for *C. elegans*, adult male and female isopods were washed, and the wash solution was used to test chemoattraction in standard assays [22,23]. Post-hoc Tukey's Honest Significant Difference (HSD) tests showed that neither N2 nor PB306 had a statistically significant attractive or repulsive behavior towards any isopod wash (Fig 1, S2 Fig, S1 and S2 Tables). However, both strains were repelled by the control repellent 1-octanol and attracted to the control attractant isoamyl alcohol [22, S1 Fig, S1 and

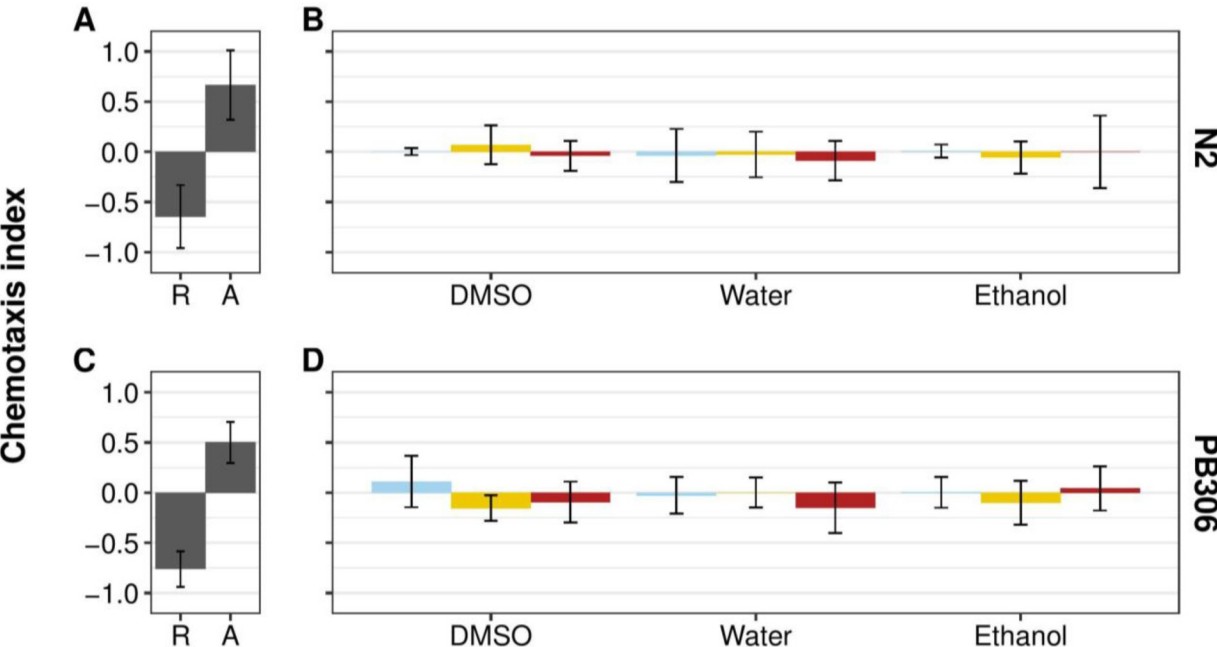

**Fig 1. *C. elegans* strains N2 and PB306 adults do not respond to *P. scaber* washes with three solvents. (A)** and **(C)** Plot of repulsive (R) and attractive (A) reference control chemotactic indices for the *C. elegans* N2 and PB306 strains. Undiluted 1-octanol was used as the repulsive negative control (left) and isoamyl alcohol diluted to $10^{-3}$ in ethanol was used as the attractive positive control (right). **(B)** and **(D)** Plot of chemotactic indices of N2 and PB306 adult *C. elegans* to undiluted *P. scaber* washes compared to neutral control references. Blue bars represent the neutral control of solvent alone. Yellow bars represent chemotactic indices to washes that were prepared from male isopods. Red bars represent chemotactic indices to washes that were prepared from female isopods. From left to right, each grouping of bars represents dimethyl sulfoxide (DMSO), deionized water, and ethanol washes (n = 40–50 animals per condition, per replicate). Error bars are standard deviation. No isopod wash was significantly different from its corresponding neutral control (see S5 Table for *p* values).

S2 Tables]. Furthermore, we found no significant differences between male or female isopod washes, indicating a lack of dependence on the sex of any chemical cocktail.

It is possible that chemoattractants from *P. scaber* are not secreted or solubilized by washes of the isopod surface. For example, fecal matter could contain a chemoattractant that is either not captured or is present in such low density in our wash methodology that it fails to elicit a chemotaxis response. Extractions of physically disrupted, ground whole animals should contain compounds that could be attractive to *C. elegans*. To capture these compounds and test this hypothesis, a whole isopod body was ground into each of the three solvents (DMSO, ethanol, and water) after washing as previously described to make a heterogeneous mixture. Like the previous washes, the N2 strain did not have a statistically significant attraction or repulsion to extractions made with any solvent (S2 Fig, S1 Table). Because no significant effects of extractions were observed, we did not test the PB306 strain.

## *C. elegans* dauers are not attracted to compounds washed from the terrestrial isopod *P. scaber*

Our results suggest that *C. elegans* adults are not attracted to compounds from *P. scaber* in laboratory chemotaxis assays. However, the life stage most commonly found in association with phoretic vectors is the dauer juvenile. This developmentally arrested larval stage is analogous to the 'infective juvenile' stage in entomopathogenic nematodes [24,25]. As the name implies, this stage has host-seeking behavior, suggesting the possibility that the *C. elegans* dauer stage is more likely to seek phoretic interactions than adults. Therefore, we tested the hypothesis that dauer individuals are attracted to chemical compounds washed from *P. scaber* in the standard chemotaxis assays used for the adult stage (Fig 2, S4 Fig, S3 Table). Post-hoc Tukey's HSD tests

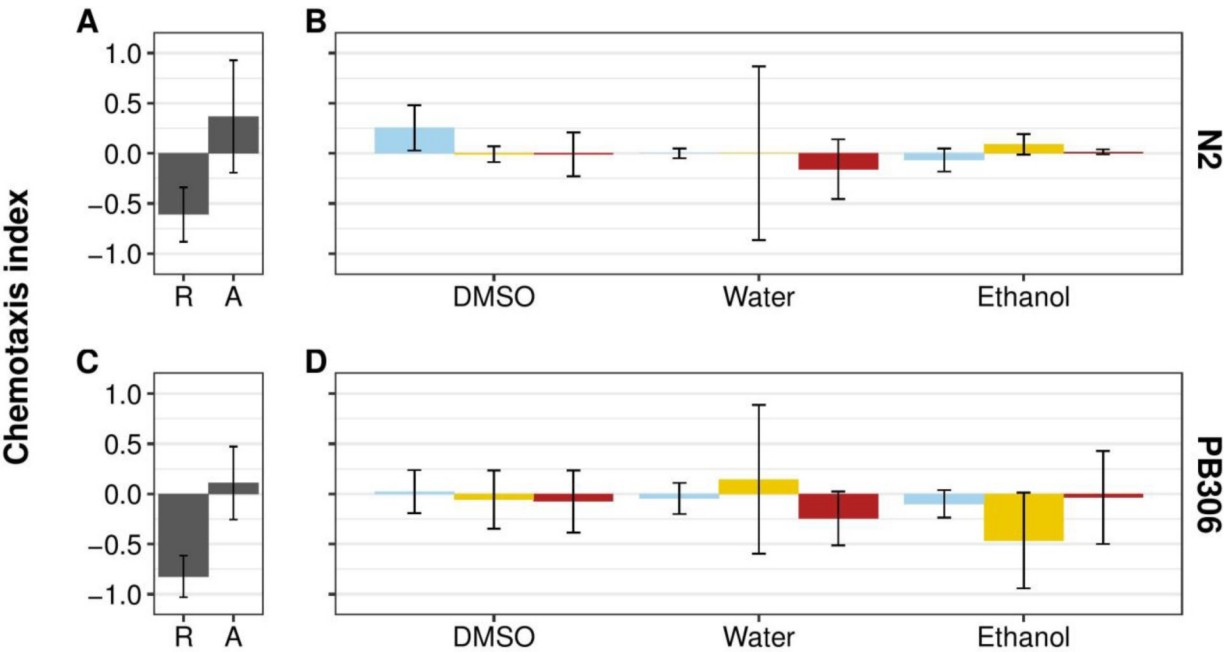

**Fig 2. *C. elegans* N2 and PB306 dauer animals do not respond to *P. scaber* washes with three different solvents. (A)** and **(C)** Plot of undiluted 1-octanol repulsive (R) and isoamyl alcohol diluted to $10^{-3}$ in ethanol attractive (A) reference control chemotactic indices for the N2 and PB306 dauers. **(B)** and **(D)** Plot of chemotactic indices of N2 and PB306 dauer *C. elegans* to undiluted *P. scaber* washes compared to neutral control references. Blue bars represent the neutral control of solvent alone. Yellow bars represent chemotactic indices to washes that were prepared from male isopods. Red bars represent chemotactic indices to washes that were prepared from female isopods. From left to right, each grouping of bars represents dimethyl sulfoxide (DMSO), deionized water, and ethanol washes (n = 40–50 animals per condition, per replicate). Error bars are standard deviation. No isopod wash was significantly different from its corresponding neutral control (see S5 Table for *p* values).

showed that both N2 and PB306 dauers did not have statistically significant attractive or repulsive behaviors toward any isopod wash, suggesting that responses to *P. scaber* do not vary based on the developmental stage of *C. elegans*. It is possible, however, that dauer animals respond differently in a manner dependent on the amount of time spent in the dauer stage but we are not aware of any literature addressing this point in *C. elegans*. Moreover, the variance among dauer individuals appears to be higher than for adults in the corresponding assay. This result was not directly tested because we found no clear attraction to *P. scaber*. The increase in variance could be caused by stage-specific differences in behaviors between dauers and adults in the absence of food, as was found previously [26,27,28].

## *C. elegans* adults are not attracted to *P. scaber* odorants

Rather than being attracted to compounds found on the body of a host, some parasitic nematode species are attracted to gaseous components of odorants, such as carbon dioxide ($CO_2$), secreted by host invertebrates [29]. Moreover, both specialist and generalist entomopathogenic nematodes responded to $CO_2$, suggesting attraction to volatile odorants could be common. To test this hypothesis using adult *C. elegans*, we adapted a gas assay [29] to measure chemotaxis in response to volatile odorants from *P. scaber* (Fig 3, S4 Table). The N2 strain of *C. elegans* was not significantly attracted to *P. scaber* odorants (mean CI = 0.07), and strain PB306 was weakly repulsed (mean CI = -0.15, p = 0.15). Overall, *P. scaber* odorants were not an attractant for either *C. elegans* strain to seek isopods.

## Discussion

Using chemotaxis assays, our results demonstrate that *C. elegans* is neither attracted to nor repulsed by chemical cues from the isopod *P. scaber* in laboratory-based chemotaxis assays. These results suggest that *C. elegans* phoresy might not be directed toward the terrestrial isopod *P. scaber* but is instead opportunistic. Consistent with this hypothesis, Lee *et al*. [4] showed that the dauer-specific behavior, nictation, is both an opportunistic behavior towards a phoretic host and is necessary for dauer individuals to disperse via the fruit fly *Drosophila melanogaster* (see [5]) for a discussion of the genetic basis underlying variation in this behavior in natural populations). However, this result does not preclude the possibility of chemoattraction to preferred hosts at other life stages but did demonstrate that for dauer individuals dispersal only happened when nictation behavior was present.

The isopods used in this study were collected from the wild and then grown in the laboratory in a controlled and sterile environment. It is possible that the change in environment could alter signals affecting chemoattraction in the wild. Our general assumption, that PB306 would likely show an attraction to isopods, was clearly not supported in this study. However, if the relevant cues come from the natural environment then it is not surprising that, when a wild strain is placed into the unnatural and sterile laboratory setting, the behavior would change as well. Additionally, given that PB306 did not display the expected behavior, we chose not to test it for chemoattraction to whole isopod extracts. Moreover, other factors besides odors and chemicals could play a role. For example, some nematode ascarosides promote aggregation among individuals and sex pheromones act as attractants [30,31,32].

In Europe, *C. elegans* has been isolated from habitats shared with *Caenorhabditis briggsae*, including co-isolation on arthropod and mollusk hosts [7]. In this case, it appears the primary ecological difference between these species is optimal temperature, creating a temporal rhythm where *C. briggsae* dominates when temperatures are warmer and *C. elegans* dominates when temperatures are cooler [7,33]. Given that *C. briggsae* has repeatedly been sampled from a variety of mollusk hosts, this result suggests that it is possible that at least some *C. elegans* could

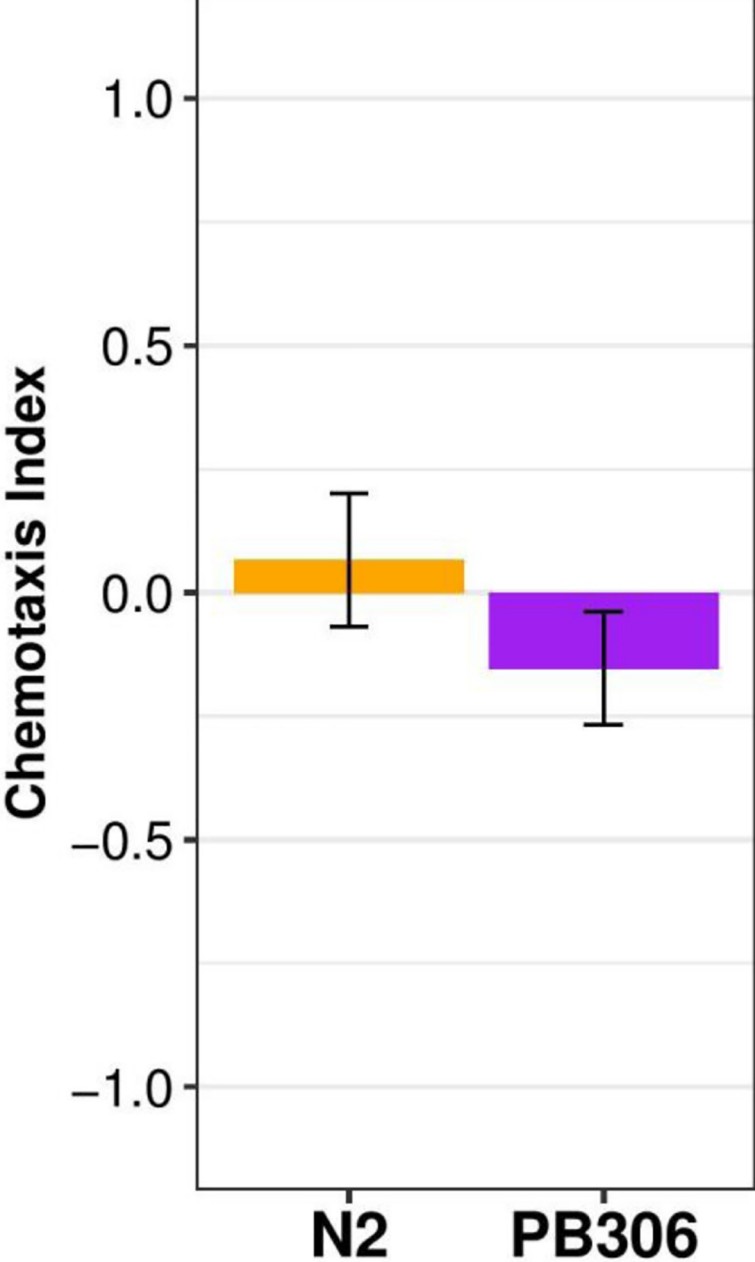

**Fig 3. *C. elegans* N2 and PB306 strains do not respond to *P. scaber* gas secretions.** The orange bar (left) represents the chemotactic indices of N2 adults to *P. scaber* gas secretions from at least three independent experiments. The purple bar (right) represents the chemotactic indices of PB306 adults to *P. scaber* gas secretions from at least three independent experiments (see S5 Table for *p* values, n = 50–100 animals per replicate). Error bars are standard deviation.

share an attraction to mollusks. Alternatively, *C. briggsae* could be attracted to hosts and *C. elegans* just follows that species. *P. hermaphrodita* shares habitats and mollusk hosts with *C. elegans* and *C. briggsae* [11,6,3] and has been shown to chemotax toward the mucus, faeces, and volatile odorants of slugs and, in the case of snails, hyaluronic acid [34,35,36,37,38]. This observation suggests that *C. elegans* might prefer mollusks to isopods such as *P. scaber* and testing attraction to these species is a good future step.

## Materials and methods

### Nematode strains

The *C. elegans* strains PB306 and N2 were used in this study. Strain data including isolation location and isotype information are available from the CeNDR website (https://www.elegansvariation.org) [39].

### Bacterial strains, growth conditions, and media

Nematodes were grown at 20˚C on modified nematode growth media (NGMA) containing 1% agar and 0.7% agarose to prevent burrowing [19] and fed the *Escherichia coli* strain OP50 according to standard methods. Plate-based chemotaxis assays were performed on 6 cm unseeded NGMA plates. Gas chemotaxis assays were performed on 10 cm unseeded NGMA plates.

### Dauer induction

Dauers were induced by bleaching gravid adults according to standard methods using K medium in place of 1X M9 solution two days before assays [40]. Embryos were titered and added to 1 mL K medium with *E. coli* HB101 lysate (5 g/1 L) and synthetic pheromone mix (5 mL/1 L) [41,42]. After growing for 48 hours at 25ºC, dauers were identified morphologically by their dark intestines and radially constricted bodies.

### Isopod culture

*Porcellio scaber* isopods were ordered as needed from www.bugsincyberspace.com. Upon each order, isopods were collected from nature just prior to being shipped and therefore do not harbor any lab adaptation. Isopods were confirmed as *P. scaber* by the animals' dull as opposed to waxy appearance, and sexed by immobilizing the animals with a short stream of carbon dioxide and examining the abdomen for key sex differences [43]. No additional selection beyond verification of sex characteristics was done. Isopods were then placed into cultures based on methods from Bhella *et al.* [44]. In brief, cultures were prepared by cutting a one-inch hole in the lid of a plastic chamber, filling the chamber with deionized water, and threading a Kimtech paper towel through the lid of the water-filled chamber and into a one-inch hole in the bottom of a smaller plastic container placed on top. This paper towel was used to wick moisture into and line the bottom of the smaller plastic container before filling the container halfway with dirt sterilized by autoclaving. Dried and fallen leaves from Elm and Oak trees were collected. They were autoclaved, ground by hand, and then added to the isopod containers as food, twice per week. The containers were covered with a lid with air holes. Isopods were divided into male and female chambers to prevent unwanted mating.

### Isopod washes and extractions

Isopods of the appropriate sex were sorted into individual 1 mL microcentrifuge tubes, with one animal per tube. 100 µL of the desired solvent (ethanol, dimethyl sulfoxide, or deionized water) was then added to the tube, and washes were prepared by washing the isopods in the solvent for 30 minutes with the tubes rotated on a mutator. In the case of extractions, at the end of the 30 minutes, animals were centrifuged at 3000 RPM for 30 seconds then ground into the solvent in the microcentrifuge tube using a small pestle. One isopod was washed per 100 µL of solvent for an undiluted (1X) concentration. Other concentrations were prepared by diluting the 1X stock with the appropriate solvent. Because all tested dilutions did not affect chemotaxis, figures in the main text depict results for the 1:1000 dilution only as this

concentration had the smallest variance. New isopod test washes and extractions were prepared for every assay.

## Chemotaxis assay

The chemotaxis assay was adapted from Margie *et al*. [23] and Bargmann *et al*. [22]. Assays were performed on unseeded 6 cm plates [45]. Plates were prepared by applying a mask with a 0.5 cm circular center origin and dividing into four quadrants. Each quadrant contained a point labeled as either test or control. See https://www.jove.com/video/50069/c-elegans-chemotaxis-assay for a step-by-step visual guide and overview of the experimental methodology. For chemotaxis assays of adults (N2 or PB306), 40–50 animals from a synchronized population were pipetted onto the center origin of the plate using a non-stick plastic pipette tip. Immediately following, one microliter of test compound, either wash or extraction, was added to opposing quadrants on the point labeled "test", and 1 μL of solvent was added to the remaining two quadrants labeled "control." One microliter of 0.5 M sodium azide was pipetted into each of the quadrants as an anesthetic to immobilize the animals. Positive and negative control plates were also prepared, using isoamyl alcohol diluted 1:1000 in ethanol and 1-octanol, respectively, as the test compounds. After worms and compounds were added to the plates, lids were replaced and a timer was set for one hour during which plates were left at room temperature undisturbed. Plates were either scored by hand immediately after the one hour incubation or they were stored at 4°C and scored later the same day according to the following method:

*Chemotaxis Index*
$$= (Total\#Animals\ in\ Test\ Quadrants - Total\#Animals\ in\ Control\ Quadrants)/(Total\#Scored\ Animals)$$

Animals that remained in the center origin, within 1 mm of the origin or the edges of the plates, or on the edges of the plates were not scored. A +1.0 chemotactic index score indicates maximum attraction to the test compound, and an index of -1.0 represents maximum repulsion.

## Gas assay

These assays were adapted from Dillman *et al*. [29]. Assays were performed on unseeded 10-cm standard NGM plates. Plates were divided into halves labeled test and control. Each half contained a point marked 1 cm from the edge of the plate along the axis line that would divide the plate into quadrants. A hole was drilled in the plate lid above each point. For each hole, flexible PVC tubing was attached and connected to a 50 mL syringe. The control syringe was filled with room air, the test syringe contained six live adult *P. scaber* animals. 50–100 adult *C. elegans* of the appropriate strain (N2 or PB306) were pipetted into the center origin with a non-stick pipette tip, covers were replaced, and the syringes were depressed at a rate of 0.5 mL/min for 60 minutes with a Harvard Apparatus Pump 22 syringe pump. Plates were then scored according to the following:

*Chemotaxis index*
$$= (Animals\ in\ Test\ Half - Animals\ in\ Control\ Half)/(Total\ Number\ of\ Animals)$$

Animals that remained in the center origin, within 1 mm of the origin or the edges of the plates, or on the edges of the plates were not scored. A +1.0 chemotaxis index score indicates maximum attraction to the test gas, and an index of -1.0 represents maximum repulsion.

### Statistics and plotting

Assays were performed in triplicate for every combination of nematode strain, isopod sex, and solvent concentration or gas. Mean chemotaxis indices (CI) were reported. Tukey HSD tests were performed comparing data from the test and neutral control plates. T-tests were performed to assess differences between a chemotactic index of zero and the relevant overall strain chemotactic index. All statistics presented in S5 Table.

## Supporting information

**S1 Fig. Isoamyl alcohol (diluted to 1:1000 in ethanol) and 1-octanol serve as control attractant and control repellant, respectively, for both the N2 (left) and PB306 (right) *C. elegans* strains.** Both strains respond neutrally to the three solvents used for the isopod washes (S1 and S2 Tables). Error bars are standard deviation.
(DOCX)

**S2 Fig. C. elegans N2 and PB306 adults respond neutrally to P. scaber washes at four different dilutions of the initial wash (1, 1:10, 1:100, and 1:1000).** Yellow bars represent washes prepared from male isopods. Red bars represent washes prepared from female isopods. Chemotactic indices from all isopod washes were not significantly different from the chemotactic indices of its corresponding neutral control (S1 and S2 Tables). Significance scores (p values) are in S5 Table. Error bars are standard deviation.
(DOCX)

**S3 Fig. C. elegans N2 adults respond neutrally to P. scaber extractions at four different dilutions of the initial extraction (1, 1:10, 1:100, and 1:1000).** Yellow bars represent extractions prepared from male isopods. Red bars represent extractions prepared from female isopods. Chemotactic indices from all isopod extractions were not significantly different from the chemotactic indices of its corresponding neutral control (S1 Table). Significance scores (p values) are in S5 Table. Error bars are standard deviation.
(DOCX)

**S4 Fig. C. elegans N2 and PB306 dauers respond neutrally to P. scaber washes at four different concentrations (1, 1:10, 1:100, and 1:1000).** Yellow bars represent washes prepared from male isopods. Red bars represent washes prepared from female isopods. Chemotactic indices from all isopod washes were not significantly different from the chemotactic indices of its corresponding neutral control (S3 Table). Significance scores (p values) are in S5 Table. Error bars are standard deviation.
(DOCX)

**S1 Table.**
(XLSX)

**S2 Table.**
(XLSX)

**S3 Table.**
(XLSX)

**S4 Table.**
(XLSX)

**S5 Table.**
(XLSX)

## Acknowledgments

The authors would like to thank members of the Andersen lab for helpful comments on the manuscript.

## Author Contributions

**Conceptualization:** Selina Deiparine, Erik C. Andersen.

**Data curation:** Erik C. Andersen.

**Formal analysis:** Selina Deiparine, Erik C. Andersen.

**Funding acquisition:** Erik C. Andersen.

**Investigation:** Selina Deiparine, Erik C. Andersen.

**Methodology:** Selina Deiparine, Erik C. Andersen.

**Project administration:** Erik C. Andersen.

**Supervision:** Erik C. Andersen.

**Validation:** Selina Deiparine, Erik C. Andersen.

**Visualization:** Erik C. Andersen.

**Writing – original draft:** Heather Archer, Selina Deiparine, Erik C. Andersen.

**Writing – review & editing:** Heather Archer, Erik C. Andersen.

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
