## [Decision Letter · Decision Letter 0]

22 Apr 2020

PONE-D-20-06744

Caenorhabditis elegans nematodes are not attracted to the terrestrial isopod Porcellio scaber

PLOS ONE

Dear Dr. Andersen,

Thank you for submitting your manuscript to PLOS ONE. After careful consideration, we feel that it has merit but does not fully meet PLOS ONE’s publication criteria as it currently stands. Therefore, we invite you to submit a revised version of the manuscript that addresses the points raised during the review process.

We would appreciate receiving your revised manuscript by July 31, 2020. To enhance the reproducibility of your results, we recommend that if applicable you deposit your laboratory protocols in protocols.io, where a protocol can be assigned its own identifier (DOI) such that it can be cited independently in the future. For instructions see: http://journals.plos.org/plosone/s/submission-guidelines#loc-laboratory-protocols

We look forward to receiving your revised manuscript.

Kind regards,

Myon-Hee Lee, Ph.D

Academic Editor

PLOS ONE

Journal Requirements:

Additional Editor Comments (if provided):

Reviewers' comments:

Reviewer's Responses to Questions

**Comments to the Author**

1. Is the manuscript technically sound, and do the data support the conclusions?

Reviewer #1: Partly

Reviewer #2: Yes

2. Has the statistical analysis been performed appropriately and rigorously? 

Reviewer #1: Yes

Reviewer #2: Yes

3. Have the authors made all data underlying the findings in their manuscript fully available?

Reviewer #1: Yes

Reviewer #2: Yes

4. Is the manuscript presented in an intelligible fashion and written in standard English?

Reviewer #1: Yes

Reviewer #2: Yes

5. Review Comments to the Author

Reviewer #1: The manuscript by Archer et al. explores the relationship between the nematode Caenorhabditis elegans and the isopod Porcellio scaber. This study investigates if the nematode is (specifically) attracted to odors and compounds present on the isopod, possibly important for phoresy. The nematode-phoront relationship is studied for two wild isolates of C. elegans: the labstrain N2 and the wild-strain PB306 – that has been isolated from a P. scaber isopod. Choice assays between isopod extractions and diverse control solvents are used to investigate if C. elegans is attracted to the isopods odors. This study provides an interesting perspective on the relation between C. elegans and its phoronts and gives insights in natural processes that could be important in the (local) spread of C. elegans nematodes.

Major points:

• This study shows a negative, which we think is not a problem and actually enriching to the literature. But, it is hard to prove a negative. Hence, we would suggest altering the title and the tone at some places in the manuscript. Actually, we think the running title is better. Given that C. elegans nematodes in the wild have been repeatedly found associated with isopods both in literature and as listed on Cendr; we think it is acceptable to trust that an association can exist. The problem with proving a negative should become explicitly clear in the discussion (again, no reason not to publish this paper; but it’s important to stress the particular conditions that this study was executed in). In the detailed comments some suggestions are listed.

• The paper suggests that C. elegans nematodes are not specifically attracted to P. scaber isopods, however we think this cannot be concluded from the use of chemical isolates only. Other factors besides odors and chemicals (possibly also unknown ones) could play a role in host seeking behavior. Therefore, limitations of the method used to investigate attraction should be critically discussed and this discussion should be reflected throughout the paper text.

• Although the authors clearly introduce that N2 is a lab-adapted strain that may have lost specific host-seeking behavior, they test the preference for whole-isopod extracts only in this strain. As such conclusions cannot be generalized to other wild-type genetic backgrounds (such as PB306).

• None of the figures shows or mentions replicates (biological or technical). Although it is deducible from the supplemental tables that the experiments were replicated properly, the numbers tested should be mentioned in the text as well.

• The choice for presenting the 1/1,000 dilution as the main result is not clearly substantiated, going back-and-forth between the supplementary figures and the main figures, this was confusing.

Detailed comments:

• It would have been easier if authors had included page numbers and line numbers for making comments.

Introduction

• The second sentence (first paragraph) is strangely formulated and confusing.

• The third sentence (first paragraph): should it not be ‘... distance by their...’?

• The sentence describing the paper by Petersen et al 2015 seems to suggest that they studied chemo attractants of slugs and snails, because of the connection to the previous sentence, but this is not the case.

• Last sentence second paragraph: ‘, other life stages have been isolated as well..’

Results

• Page 11 ‘as such it is likely to have invertebrate’: change ‘it’ to ‘PB306’ to prevent confusion.

• Page 11 ‘...to hypothesize that these strains...’: change ‘these strains’ to ‘N2 and PB306’

• Page 11 ‘nonpolar compounds were used’

• Page 15 describes that PB306 was weakly repulsed, but no test for significance or p-value is mentioned.

Figures

• A general method figure would help understanding the experiments performed more easily.

• Fig 2/Fig S4 The variation in dauer larvae choice assays appears to be larger than the variation measured for adults. Could the authors discuss or explain the difference? Moreover, it would be good to show the variation in control experiments like for Fig S1.

Discussion

• The first paragraph of discussion does not seem to link previous findings to this study. Instead it discusses previous findings (by Lee et al 2012 and 2017) without clear connection to the current findings.

• Where does the observation that dauers are most commonly associated with invertebrates without a chemical cue come from? No references were added to that statement. Also, if more is known about chemical cues of vectors in general it would be interesting to discuss these. Or mention that little is known.

• What type of chemicals could be attractants for nematodes? Is anything known about their chemical composition?

• The origin of the nematodes is clearly discussed, but not that of the isopods. Could they be adapted to the lab themselves? Additionally, perhaps these isopods are smelled in natural situations by the nematodes because of the organisms they associate with or produce different cues because of the (fresh) food they eat. Could it be that lab-grown sterile animals lack scent? The potential effects of using isopods in a laboratory setting should be discussed.

• The study uses dauers 48h after induction; however from other nematode species (e.g. plant-parasitic cyst nematodes) it is known that older dauers react differently to environmental cues. It is possible that the same is true for C. elegans (although I’m not aware of literature investigating this).

Material and methods

• ‘PB306 and N2.’ This is not a sentence.

• P. scaber natural origin is not described. Are these wild or lab animals? Where do they occur? Global or local?

• Unclear if there is any age synchronization or selection criterium for the isopods selected for the experiments.

Reviewer #2: The authors describe some experiments to test if C. elegans is chemically attracted to the isopod P. scaber, given elegans has been collected from P. scaber in the past, and can use P. scaber for dispersal.

Two nematode genotypes are tested, the lab adapted N2 reference and a wild isolate collected from isopods, PB306, as adults and as dauer larvae. Isopods are tested by sex. Standard plate-based attraction assays are used, testing volatiles from live isopods, and polar and non-polar extracts.

The experiments are well described and sufficiently replicated, and the results are decisively against any strong, unconditional attraction, which is the primary hypothesis of interest.

I'm not convinced the statistical analysis makes full use of the data, but it is clear that the main conclusion will not be sensitive to method. Whether there are significant effects of genotype (PB306 appears to be generally more repulsed than N2, perhaps), or of the many other tested factors, would be better answered by a joint (e.g., binomial linear model using the raw counts), rather than pairwise, analysis.

I have made some minor comments on interpretation and communication in the commented pdf.

6. PLOS authors have the option to publish the peer review history of their article (what does this mean?). If published, this will include your full peer review and any attached files.

Reviewer #1: Yes: Mark G. Sterken and Lisa van Sluijs

Reviewer #2: No

---

## [Author Response · Author response to Decision Letter 0]

17 May 2020

Our responses to the reviewers' comments are included as a PDF.

---

## [Decision Letter · Decision Letter 1]

8 Jun 2020

The nematode Caenorhabditis elegans and the terrestrial isopod Porcellio scaber likely interact opportunistically

PONE-D-20-06744R1

Dear Dr. Andersen,

We’re pleased to inform you that your manuscript has been judged scientifically suitable for publication and will be formally accepted for publication once it meets all outstanding technical requirements.

Kind regards,

Myon-Hee Lee, Ph.D

Academic Editor

PLOS ONE

Additional Editor Comments (optional):

Reviewers' comments:

Reviewer's Responses to Questions

**Comments to the Author**

1. If the authors have adequately addressed your comments raised in a previous round of review and you feel that this manuscript is now acceptable for publication, you may indicate that here to bypass the “Comments to the Author” section, enter your conflict of interest statement in the “Confidential to Editor” section, and submit your "Accept" recommendation.

Reviewer #1: All comments have been addressed

2. Is the manuscript technically sound, and do the data support the conclusions?

Reviewer #1: Yes

3. Has the statistical analysis been performed appropriately and rigorously? 

Reviewer #1: Yes

4. Have the authors made all data underlying the findings in their manuscript fully available?

Reviewer #1: Yes

5. Is the manuscript presented in an intelligible fashion and written in standard English?

Reviewer #1: Yes

6. Review Comments to the Author

Reviewer #1: We want to thank the authors for addressing our concerns and questions. Based on this version we have no additional questions and comments.

7. PLOS authors have the option to publish the peer review history of their article (what does this mean?). If published, this will include your full peer review and any attached files.

Reviewer #1: Yes: Mark Sterken and Lisa van Sluijs

---

## [Editor Report · Acceptance letter]

9 Jun 2020

PONE-D-20-06744R1 

The nematode *Caenorhabditis elegans* and the terrestrial isopod *Porcellio scaber* likely interact opportunistically 

Dear Dr. Andersen:

I'm pleased to inform you that your manuscript has been deemed suitable for publication in PLOS ONE. Congratulations! Your manuscript is now with our production department. 

Kind regards, 

on behalf of

Dr. Myon-Hee Lee 

Academic Editor

PLOS ONE